

# Systematic review of marine environmental DNA metabarcoding studies: toward best practices for data usability and accessibility

Meghan M. Shea[1], Jacob Kuppermann[2], Megan P. Rogers[3], Dustin Summer Smith[2], Paul Edwards[4] and Alexandria B. Boehm[5]

[1] Emmett Interdisciplinary Program in Environment & Resources (E-IPER), Stanford University, Stanford, CA, United States of America
[2] Earth Systems Program, Stanford University, Stanford, CA, United States of America
[3] Program in Human Biology, Stanford University, Stanford, CA, United States of America
[4] Program in Science, Technology and Society, Stanford University, Stanford, CA, United States of America
[5] Department of Civil and Environmental Engineering, Stanford University, Stanford, CA, United States of America

Corresponding author
Meghan M. Shea,
mshea@stanford.edu

## ABSTRACT

The emerging field of environmental DNA (eDNA) research lacks universal guidelines for ensuring data produced are FAIR–findable, accessible, interoperable, and reusable–despite growing awareness of the importance of such practices. In order to better understand these data usability challenges, we systematically reviewed 60 peer reviewed articles conducting a specific subset of eDNA research: metabarcoding studies in marine environments. For each article, we characterized approximately 90 features across several categories: general article attributes and topics, methodological choices, types of metadata included, and availability and storage of sequence data. Analyzing these characteristics, we identified several barriers to data accessibility, including a lack of common context and vocabulary across the articles, missing metadata, supplementary information limitations, and a concentration of both sample collection and analysis in the United States. While some of these barriers require significant effort to address, we also found many instances where small choices made by authors and journals could have an outsized influence on the discoverability and reusability of data. Promisingly, articles also showed consistency and creativity in data storage choices as well as a strong trend toward open access publishing. Our analysis underscores the need to think critically about data accessibility and usability as marine eDNA metabarcoding studies, and eDNA projects more broadly, continue to proliferate.

# INTRODUCTION

## Contextualizing eDNA metabarcoding research

Human activities are increasingly threatening ecosystems around the globe—from terrestrial (*Tilman et al., 2017*) to freshwater (*Reid et al., 2019*) to marine (*Halpern et*

*al., 2019*) environments—impacts that have implications for ecological and human communities alike (*Pecl et al., 2017*). In this era of global environmental change, combining diverse datasets to understand how ecosystems are being altered is increasingly vital (*Henry et al., 2008*; *Schmeller et al., 2017*). Yet large spatial, temporal, and taxonomic gaps remain in biodiversity data, a challenge that eDNA can help address (*Altermatt et al., 2020*). eDNA refers to genetic material found in an environmental sample (*Taberlet et al., 2012*)—including from terrestrial environments (*Hassan et al., 2022*), water (*Rees et al., 2014*; *Valentini et al., 2016*), and more recently, even air (see *Clare et al., 2021*; *Lynggaard et al., 2022*). Sometimes, this complex mix of intracellular and extracellular DNA is used to detect whether particular organisms are present, often using species-specific primers and quantitative PCR (*Taberlet et al., 2018*). In other cases, researchers want to characterize biodiversity more generally, utilizing broader primers and next-generation sequencing to identify many taxa from the same sample, an approach known as metabarcoding (*Taberlet et al., 2012*).

eDNA metabarcoding has many potential advantages over conventional biodiversity monitoring; it can be less expensive (*Evans et al., 2017*; *Hering et al., 2018*) and less invasive (*Beja-Pereira et al., 2009*), does not require in-depth taxonomic knowledge in the field (*Hering et al., 2018*), and provides comparable, or complementary, species identification to approaches based on morphotaxonomy (*Fediajevaite et al., 2021*; *Keck et al., 2022*; *McElroy et al., 2020*). Because collecting eDNA samples can be easier than other biodiversity monitoring techniques, such as trawls and visual surveys, there is even a burgeoning interest in using eDNA tools with non-experts, a potential avenue for expanding the scale of biodiversity data collection (*Deiner et al., 2017*; *Meyer et al., 2021*; *Miya et al., 2022*). Especially in marine environments, where biomonitoring can be particularly challenging, eDNA metabarcoding has been used for myriad purposes (*Miya, 2022*). Researchers have employed metabarcoding to monitor many types of organisms, including threatened (*Nester et al., 2020*; *Nichols & Marko, 2019*; *Truelove, Andruszkiewicz & Block, 2019*), cryptic (*Bessey et al., 2020*; *Parsons et al., 2018*; *Port et al., 2016*; *Thomsen et al., 2012*), commercially fished (*Russo et al., 2020*; *Thomsen et al., 2016*), or invasive (*Ardura et al., 2015*; *Borrell et al., 2017*; *Von Ammon et al., 2019*; *Westfall, Therriault & Abbott, 2020*) species. eDNA metabarcoding projects have also studied how biodiversity is influenced by environmental conditions (*Closek et al., 2019*; *Djurhuus et al., 2020*; *Ghosh & Bhadury, 2018*), anthropogenic impacts (*Andriyono, Alam & Kim, 2019*; *Bakker et al., 2017*; *Boussarie et al., 2018*; *Clementi et al., 2021*; *Cordier et al., 2019*; *DiBattista et al., 2020*; *Kelly et al., 2016*), and interventions such as marine protected areas (*Gold et al., 2021*). Results from eDNA studies have even been used to forecast future biodiversity changes (*Gallego et al., 2020*).

This proliferation of eDNA metabarcoding studies has increasingly allowed published data to be reused and reanalyzed to answer new questions, particularly through meta-analyses, *i.e.,* systematic studies which merge the findings from individual articles and statistically analyze them to calculate an overall effect or trend. In the past several years, meta-analyses of eDNA metabarcoding studies have compared eDNA methods to traditional biodiversity monitoring methods (*Fediajevaite et al., 2021*; *Keck et al., 2022*; *McElroy et al., 2020*). While these studies are able to synthesize new findings across previous

research, they also highlight a major challenge to the growing eDNA field: a lack of widely accepted best practices. Several authors have identified that they had trouble integrating data across various projects in a common framework, foregrounding the importance of systematic and comparable data collection and reporting procedures (*Fediajevaite et al., 2021*; *Keck et al., 2022*).

## Existing efforts toward eDNA metabarcoding best practices

Calls to develop best practices for eDNA research are not new, and often focus on one of two related agendas: *reproducibility* and *FAIRness*. The first is concerned with whether eDNA metabarcoding studies can be replicated. From the field to the laboratory to the bioinformatics pipeline, eDNA metabarcoding rests on a long list of complex methodological decisions that add uncertainty, variability, and bias to results (*Cristescu, 2014*; *Mathieu et al., 2020*; *Zinger et al., 2019*). If these decisions are not justified and reported, it becomes challenging to replicate existing studies; if they differ across existing studies, it becomes nearly impossible to compare findings. In a recent review of terrestrial and freshwater metabarcoding studies, *Dickie et al. (2018)* found that only 5% of studies would be replicable, due to subjective or inappropriate field methods, or insufficient published methodological information. In response to reproducibility concerns, and a desire to increase confidence in eDNA for use in regulatory and management contexts, there have been many efforts to provide standardized methodological and reporting guidance. Many such efforts have drawn directly on published literature to inform their suggestions, using reviews of published studies to provide recommendations for sampling (*Dickie et al., 2018*), laboratory methods (*Lear et al., 2018*), data submission (*Tedersoo et al., 2015*), and reducing variability and uncertainty across all stages of research (*Mathieu et al., 2020*). Additional studies have proposed standardized methods for using eDNA to survey specific taxa, such as benthic macroinvertebrates (*Duarte et al., 2021*) and fish (*Shu, Ludwig & Peng, 2020*), as well as for using eDNA for particular applications, such as biosecurity surveillance (*Bowers et al., 2021*). Such proposals do not exist only in the peer-reviewed literature; many entities in a range of countries have also sought to develop standards. In North America, the Canadian Standards Association has recently published a set of minimum reporting requirements for eDNA (*CSA Group, 2021*), and the California Molecular Methods Workgroup has designed a set of standard procedures for eDNA sampling (*California Water Quality Monitoring Council, 2023*). Across Europe, the DNAqua-Net initiative has brought together academic experts and other biomonitoring stakeholders to create a roadmap for eDNA sampling, including a comprehensive methodological guide (*Blancher et al., 2022*; *Bruce et al., 2021*). In Japan, the eDNA Society has created a detailed manual, with associated Japanese-language videos, that proposes a standardized set of sampling and experimental protocols (*Minamoto et al., 2021*). While these efforts help guide researchers toward greater methodological standardization, the fact that so many parallel recommendations have been published highlights the challenge of adopting a single, universal set of practices.

At the same time, there is a parallel, and somewhat overlapping, concern about whether eDNA data are FAIR: findable, accessible, interoperable, and reusable (*Wilkinson et al.,*

*2016*). The FAIRness of eDNA data is governed by many different factors, including data storage decisions (where underlying sequence data are held) and metadata practices (what information is included with eDNA data). *Berry et al. (2021)* note that eDNA metabarcoding studies produce rich datasets of occurrences of particular organisms, but even when the underpinning data are published, they are often not formatted for reuse or integrated into existing biodiversity databases, thus limiting the potential value of those data. In a systematic review of freshwater eDNA studies, *Nicholson et al. (2020)* found that none of the metadata metrics they analyzed were included in every paper in their sample, and that the more specific metadata metrics were reported less frequently than broad metrics. In response, making eDNA data accessible has become an increasing priority in the field. For example, the 2020 International Virtual Conference on the use of Environmental DNA in Marine Environments dedicated one of its four days to keynotes and discussions about "'metadata' and participation/interoperability with data systems from other disciplines" (*POGO, 2020*). Efforts to develop metadata standards for eDNA in particular have been initiated from many different directions. One example emerging from the perspective of biodiversity databases is the Global Biodiversity Information Facility guide to publishing DNA-derived data (*Andersson et al., 2021*). Additional examples have emerged from the perspective of eDNA users, such as the CA Molecular Methods Working Group metadata reporting standards (*California Water Quality Monitoring Council, 2023*) and minimum information recommendations from the 2013 International Congress for Conservation Biology (*Goldberg et al., 2016*).

The push for reproducibility and accessibility of eDNA studies is also informed by broader calls to address data challenges in research communities that intersect with eDNA work, from disciplinary contexts such as ecology (*Gerstner et al., 2017*; *Haddaway & Verhoeven, 2015*; *Reichman, Jones & Schildhauer, 2011*) and methodological contexts such as omics (*Chervitz et al., 2011*; *Field et al., 2009*). Further, a growing number of initiatives are relevant, but not exclusive, to eDNA metabarcoding studies. For example, the Genomics Standards Consortium developed minimum information about any (x) sequence (MIxS) as a general guide for metadata included with sequence data (*Yilmaz et al., 2011*), and the Intergovernmental Oceanographic Commission's Ocean Best Practices System developed the Minimum Information for an Omic Protocol (MIOP) as a guide for ocean-specific omics research (*Samuel et al., 2021*). Recognizing that these minimum information standards are not always sufficient for ensuring that ecologically relevant metadata is captured in traditional nucleotide sequence databases, such as the National Center for Biotechnology Information's (NCBI's) Sequence Read Archive, the Genomic Observatories Metadatabase (GeOMe) was developed to host geographic and ecological metadata that could be linked with associated genetic data (*Deck et al., 2017*).

## Objectives of the review

With this ever-growing landscape of specific eDNA initiatives, and more broadly relevant platforms, how do eDNA metabarcoding projects manage data challenges in practice? While some studies have addressed this question in the context of reproducibility, far fewer have studied the accessibility of eDNA data (*e.g.*, *Nicholson et al., 2020* for freshwater), and

none have focused on marine environments. Yet, emerging technologies do not always mesh well with existing infrastructures and protocols, demonstrating that attention ought to be paid not only to how data are produced, but what happens after that production. In focus, this review elevates an understudied aspect of eDNA data challenges: FAIR data principles. In scope, it centers on marine studies, a specific field where conversations about data accessibility are already happening (*e.g.* the 2020 International Virtual Conference on the use of Environmental DNA in Marine Environments and the 2022 2nd National Workshop on Marine Environmental DNA), but where no systematic reviews have tracked current data practices. In the following systematic review, we analyze published marine eDNA metabarcoding studies, with a particular eye toward factors that impact the FAIRness of the underlying data, including metadata and data storage practices, in order to highlight challenges, as well as promising trends, in the continued quest toward usable and accessible eDNA data.

## METHODS

### Literature selection

Using standard systematic review protocols (*Moher et al., 2015*), we conducted a literature search of peer-reviewed articles indexed in Web of Science, PubMed, and Scopus (see Fig. 1). On all platforms, we used the search string *("environmental DNA" OR eDNA) AND (marine OR ocean\* OR seawater OR saltwater OR sea)* across titles, abstracts, and keywords to broadly identify articles using eDNA in marine environments, published up to 31 December 2020. Using this search strategy, we were only selecting for articles that self-identified as studying "environmental DNA", rather than articles using the same, or similar, methods. Because many eDNA articles are published in the journal *Environmental DNA*, which at the time of searching was not yet indexed in any of the databases above, we additionally searched that journal's corpus using the same search string and date range and added the returned articles to our sample. After removing duplicates using the systematic review organizational platform Covidence, 1,014 articles remained for us to screen.

We utilized a two-phase screening process, first identifying potentially relevant articles from the title and abstract, and then further investigating the full text of articles passing the initial screen. During both phases, all articles were screened by two members of the research team (MS, JK, MR, or DS); any disagreements over the relevancy of a given article in either phase were resolved with the full screening team (MS, JK, MR, and DS). Selected peer-reviewed articles met the following five criteria: (1) were published in English, (2) primarily reported novel scientific findings (no book chapters, review papers, perspective pieces, or similar), (3) collected eDNA samples directly (no modeling papers), (4) reported eDNA data from at least one water sample (no papers with samples exclusively from sediment, tissue, gut, or similar) from a marine environment (no papers with exclusively freshwater sampling), and (5) utilized a metabarcoding approach to sequence their sample(s). The title/abstract screening yielded 276 potentially relevant articles, which were narrowed to 120 relevant articles by the full-text screening. Sixty of these articles were selected for inclusion in the analysis, as detailed below.
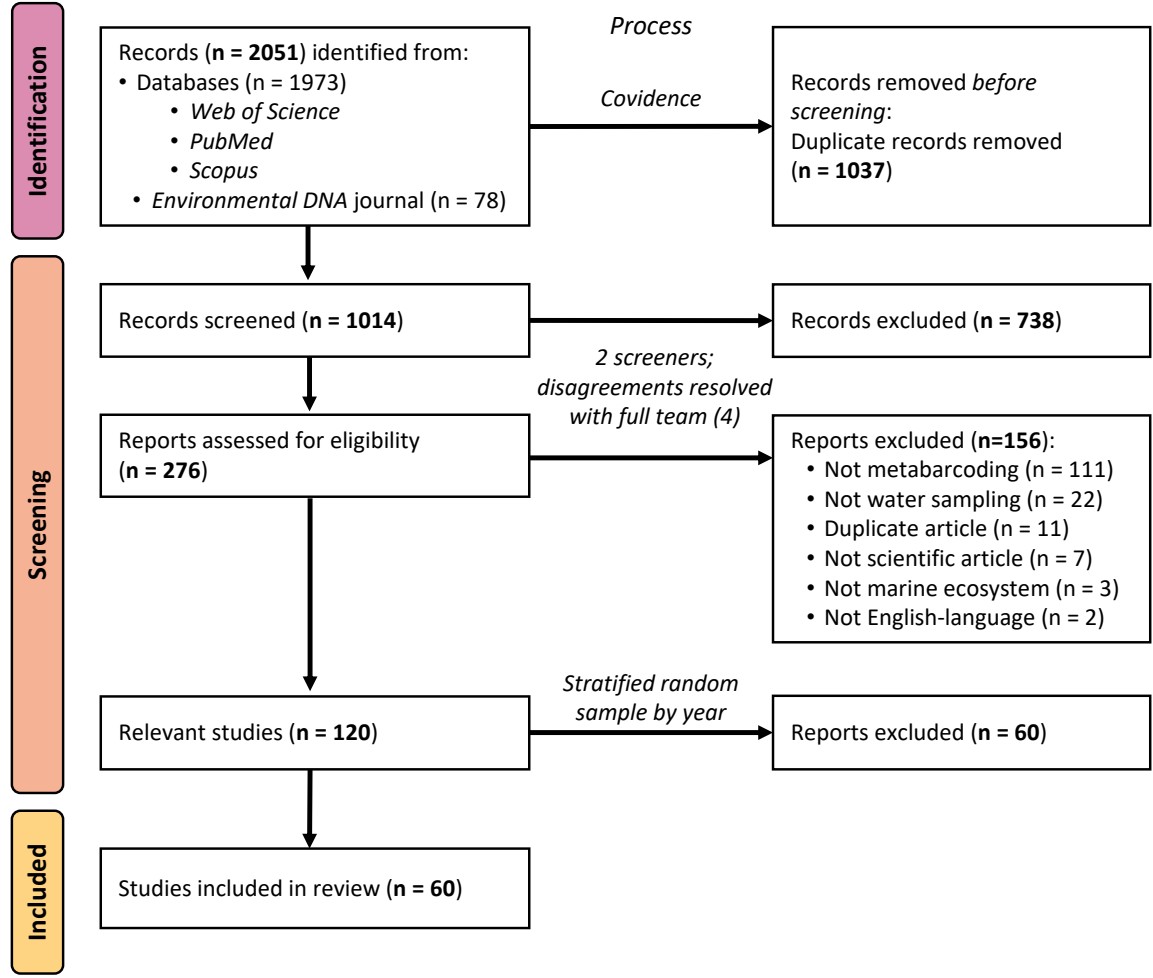

**Figure 1** **PRISMA flow diagram of systematic review process.** PRISMA flow diagram detailing the systematic review process and how many articles were included or excluded at any given step.

## Data collection

Elements to be extracted from relevant articles were developed from criteria used in a review of freshwater eDNA metadata practices (*Nicholson et al., 2020*) and expanded using criteria from existing eDNA metadata frameworks, including the Global Biodiversity Information Facility (GBIF) recommendations for metabarcoding data (*Andersson et al., 2021*), as well as the research team's knowledge of environmental DNA research. Extraction elements were clarified and refined *via* a pilot using approximately 10% of the relevant articles, with the full data collection team (MS, JK, MR, and DS) independently extracting each pilot article and discussing all differences.

Ultimately, we compiled a list of approximately 90 elements to extract, which fell into several broad categories that structure the results section below. The first set of categories helped provide overarching context. For each article, we first recorded *general article characteristics*, including basic publication information (authors, year published,

journal name, open access status) and information about the methodological scope (target taxa, metabarcoding loci used, type of environment sampled, complementary methods used beyond eDNA metabarcoding). We also extracted information about the *geographic scope* of the articles, including the institutions of first and last authors of the paper (to represent the dominant locations where the full project was conducted) and where samples were collected. For papers that gave geographic coordinates for sampling sites, sampling location was recorded as the centermost coordinate for each distinct geographic area. When specific coordinates were not given, sampling location was estimated from included maps or location information in the text. For each article, the relationship between the institution(s) and the sampling location(s) was mapped using the associated institution (first or last) with the smallest average distance to its sampling sites. Because crucial metadata and data storage information was often contained in the *supplementary information* of the articles we analyzed, we cataloged the number and type of supplementary information files associated with each article.

We extracted several elements related to *data storage*. We recorded whether articles stated that they had published their underlying sequence data (uploaded FASTQ, or similar, files as supplementary information or in an external repository) and where in the article that statement was made (in a data availability statement, in the methods section, elsewhere). If articles published underlying sequence data, we then extracted information about the platform on which the data were published and followed the link or accession number given to record whether the sequence data were indeed accessible at that location, the file format of the data, and whether the platform provided a unique citation for that dataset. In cases where the link or accession number did not lead to a valid dataset, we emailed the corresponding author(s) and asked whether they knew the data were not available as stated, and why that might have occurred.

Beyond just understanding how articles stored and cataloged their sequence data, we also wanted to understand how articles captured and recorded metadata related to the project, a category we termed *metadata inclusion*. Across all articles in the sample, we recorded whether papers included 60 different types of metadata across 13 categories (a full list of metadata elements analyzed can be found in the Results section). Importantly, we were only assessing the presence or absence of the information, making no value claim about the validity of the information included (cf. *Dickie et al., 2018*). For example: we recorded whether articles included *any* information about filter size and type, not whether articles used *particular* filter sizes and types. We then averaged the percent inclusion across the elements within each metadata category; these averages help show general trends across the different categories, but we do not intend to suggest that all metadata elements are equally important. Because of our interest in data accessibility, we also recorded additional information about two of our metadata elements, statistics and bioinformatics analysis scripts, including where the scripts were published.

Additionally, there have already been some efforts to provide metadata guidance for eDNA metabarcoding studies, so we wanted to further use our *metadata inclusion* data to assess how easily existing studies would be able to comply with new standards: that is, are studies already including the recommended information for these standards, or

would it likely be challenging for studies to adopt them. We selected one illustrative standard to study—the GBIF guide for publishing DNA-derived data (*Andersson et al., 2021*). GBIF is a global repository of biodiversity observations originally designed for traditional biodiversity sampling records: where an organism has been collected and observed, and then taxonomically identified, either visually or morphologically (*Andersson et al., 2021*). In contrast, occurrences derived from eDNA sampling involve many additional steps between the collection of material to a final list of species, steps that all necessitate additional metadata for the final occurrence to be sufficiently contextualized (*Andersson et al., 2021*). Recognizing that DNA-derived occurrences need specialized standards, GBIF released a set of additional recommended fields for submitting DNA-derived data, including separate guidance for both metabarcoding and ddPCR/qPCR (*Andersson et al., 2021*). While journals and funders sometimes require that eDNA sequence data be submitted to sequence read archives, there are rarely mandates that the associated biodiversity occurrences are submitted to repositories like GBIF. Therefore, the GBIF recommendations for metabarcoding data represent a case where the difference between what studies are already including and what the recommendations necessitate really matters; if eDNA projects do not already have the metadata on hand to upload their data to GBIF, it seems unlikely that they will. To assess this potential discrepancy, we included 13 metadata elements that corresponded with GBIF recommendations in order to see how well existing studies would be able to comply with these proposed standards. The selection of 13 elements that we chose was only a subset of the full list of GBIF recommendations. We excluded all fields that would have been identical across all studies in our sample (such as environmental medium) or were more broad than other metadata elements we investigated (such as sampling protocol).

Finally, beyond just studying the different types of metadata included in the articles, we were also interested in what studies referred to as metadata, what we termed *metadata language*. That is, we were curious as to what types of information were designated as metadata by authors. We anticipated that different articles might use the word "metadata" to refer to very different kinds of information; for example, one article could call a supplementary table of temperature and salinity records "metadata", whereas another article could use the term to describe all information needed to construct reference libraries.

Due to the comprehensive nature of these elements, we opted to fully extract half of our sample of 120 relevant articles. These were selected *via* a stratified random sample by publication year, so that the articles included would be representative of any changes in metadata or data storage practices over time. All articles were extracted by two researchers independently (of MS, JK, MR, DS); a third researcher (MS, JK, or MR) compared these extractions and resolved any differences across all elements. While the configuration of researchers extracting and resolving each article varied to reduce bias, one researcher (MS) either extracted or resolved every article in the sample to ensure consistency. Articles were analyzed using basic descriptive statistics.
## RESULTS

### General article characteristics

The 60 articles analyzed represented a range of sample types, methodological approaches, journals, and institutions (see Fig. 2). Most studies (90%) collected water samples for eDNA analysis in the field, with the additional minority of studies collecting either ballast water or samples from laboratory or mesocosm experiments. Only 2 studies (3.3%) combined these approaches, coupling field sampling with laboratory sampling. Per the selection criteria, all studies conducted eDNA metabarcoding on marine seawater samples. However, some studies extended the types of samples analyzed and the methodological approaches employed. One-quarter of studies (25%) analyzed samples beyond marine seawater, such as sediment samples, and some studies (18.3%) utilized other eDNA approaches as well, including qPCR analysis and shotgun sequencing. Additionally, almost one-third of studies (30%) employed other biodiversity monitoring methods in conjunction with eDNA sampling, such as acoustic sampling and net trawls.

The studies were published in an array of academic journals, some broad (such as *Scientific Reports* and *PLOS ONE*) and some disciplinarily specific (such as the *Journal of Molluscan Studies* and the *Journal of Fish Biology*). The sample of 60 articles collectively appeared in 30 different individual journals, with *Environmental DNA* (11 articles) and *Ecology and Evolution* (six articles) publishing the most articles in the sample and the majority of journals (63.3%) publishing only one article in the sample. Most studies (91.6%) were published between 2017 and 2020, with the earliest article in the sample appearing in 2010 (Fig. 3). The majority of articles (78.3%) were published open access.

Most articles (75%) were focused on broad ranges of taxa, while the remaining articles were focused on specific families, genera, or species (23.3%) or were unspecified (1.7%). Often, when a specific species was targeted, it was identified as invasive or non-indigenous. The articles also varied in which genetic loci they targeted *via* metabarcoding. The most used target region was COI (41.7%), followed by 18S rRNA (36.7%), 12S rRNA (33.3%), 16S rRNA (25%), and all other target regions (15%). A slight majority (56.7%) of articles targeted a single genetic locus, while the remainder used various primers to target multiple genetic loci. The most common groupings of genetic targets were COI & 18S rRNA (11.7%) and COI, 16S rRNA & 18S rRNA (5%).

### Geographic scope

Across the first and last authors of the sampled articles, 78 different institutions were represented, ranging from universities and government agencies, to specialized centers and businesses (Fig. 4A). Most institutions (76.9%) were only associated with one article, but of the remainder, Curtin University (five articles), Université Laval (four articles), and Stanford University (four articles) appeared most frequently. The institutions were located in 19 different countries: the United States (17), China (12), the United Kingdom (nine), Canada (six), Australia (five), New Zealand (five), Spain (three), France (three), Germany (three), Japan (three), South Korea (two), Switzerland (two), Ukraine (two), Colombia (one), Denmark (one), Lithuania (one), Norway (one), South Africa (one), and the Netherlands (one).

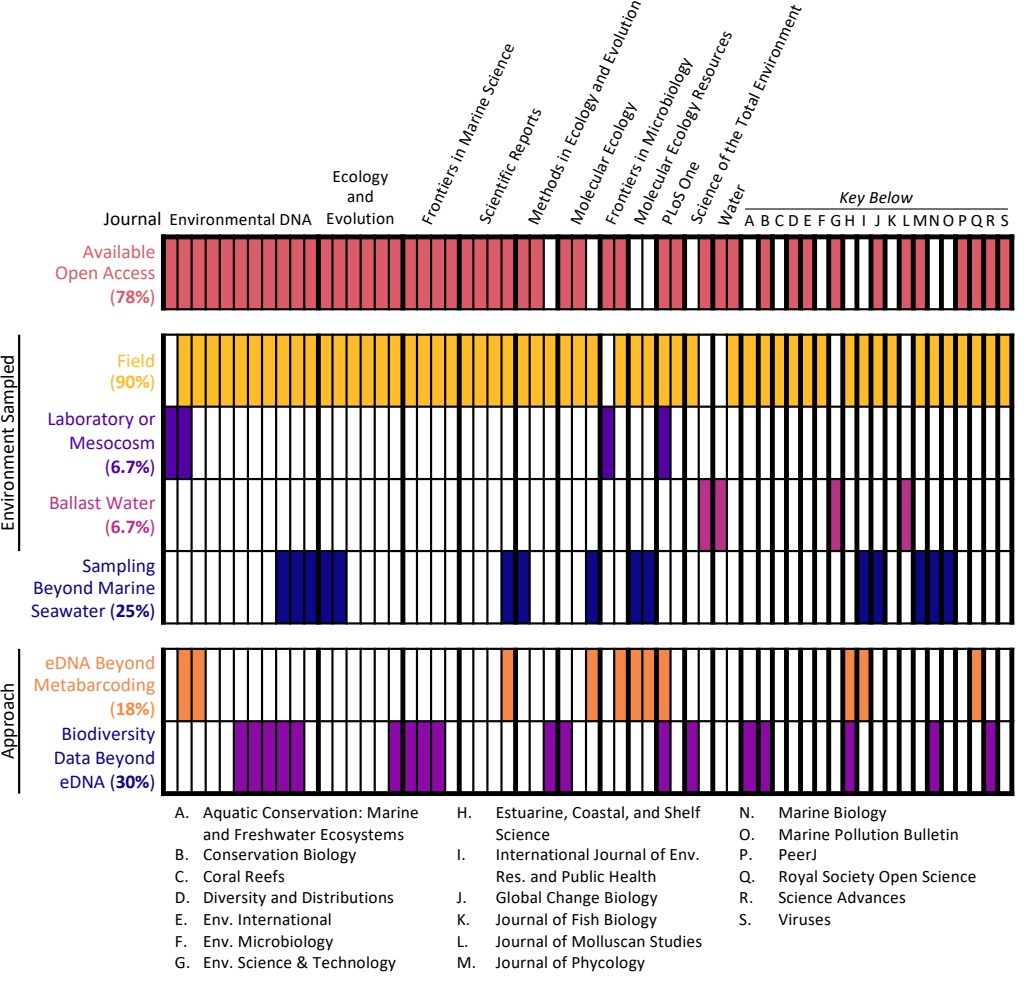

**Figure 2** **Article scope organized by journal.** Each column denotes a single article, with various characteristics of its scope: open access status, type of environment samples, whether samples other than marine seawater were collected, and whether approaches in addition to eDNA metabarcoding were used.

Across the surveyed articles, samples were collected in 31 different countries and territories, as well as international waters. The locations sampled spanned all continents save for Antarctica, though locations in Africa and South America were sampled in only one article each (Fig. 4B). Articles collected samples most frequently in the United States (14 articles), Canada (seven articles), China (six articles), Australia (four articles), Spain (four articles), and Norway (three articles). Several countries and territories contained sampling locations, but no first or last author institutions, including Jamaica, Turks & Caicos, the Bahamas, the British Virgin Islands, Belize, Poland, Saudi Arabia, Turkey, Georgia, Russia, and Ireland. 28.3% of articles sampled in multiple distinct geographical regions as described by their authors.
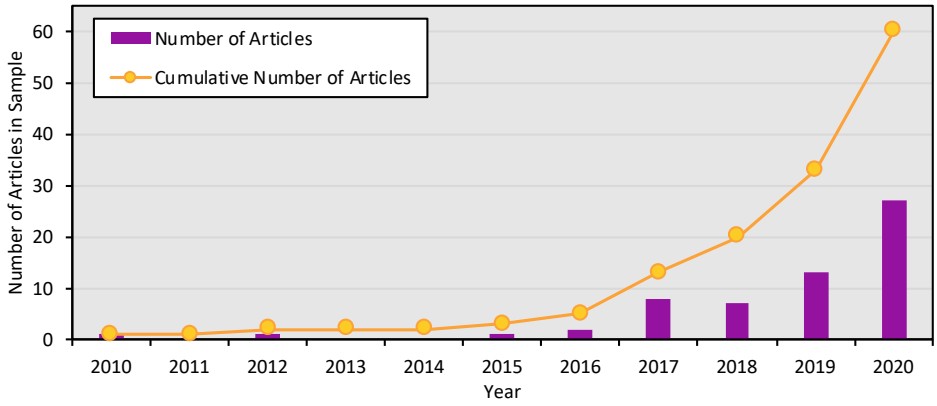

**Figure 3 Number of articles focused on marine eDNA metabarcoding by year.** Number of articles in the sample published in each year, and cumulatively, from 2010 to 2020.

## Supplementary information

Nearly all papers in the sample (93.3%) contained at least one supplementary information file, and they generally included more than one (mean: 3.9, median: 1.5). The maximum number of files associated with a single paper in our sample was 20 individual files. The most common file format for supplementary information files was Microsoft Word documents, included in just over half (58.9%) of all articles with supplements. Other common data formats included PDFs (33.9%) and Microsoft Excel spreadsheets (37.5%), both found in over one-third of articles with supplements. Typically, even if an article had multiple individual supplementary files, all files were the same file type, most commonly PDFs (15 articles) or Word documents (13 articles). In other cases, articles utilized a mixture of different file types. Every article with supplementary files utilized at least one of the most commonly found file formats—Word documents, PDFs, or Excel spreadsheets—but only one article utilized all three. Other file formats found in supplementary information sections included various image data formats, as well as .FASTA and .FASTQ files.

## Data storage

The majority of studies (76.7%) indicated that they had published sequence data, although the platform used and success of that publication (that is: whether the sequence data could actually be accessed with the information provided) varied across the studies (see Fig. 5). The most popular sequence data publication platform across articles (61.7%) was the Sequence Read Archive (SRA), a public repository for DNA sequencing data as part of the International Nucleotide Sequence Database Collaboration. The underlying SRA can be accessed through the different portals of the collaborating institutions, the National Center for Biotechnology Information (NCBI), the European Bioinformatics Institute, and the DNA Data Bank of Japan (DDBJ); across articles, it was variously referred to as the NCBI SRA, GenBank (a component of the NCBI SRA), the European Nucleotide Archive, and the DDBJ SRA. One study that published sequence data (1.7%) utilized a similar platform not synced with the above SRA, the China National GeneBank Sequence Archive.
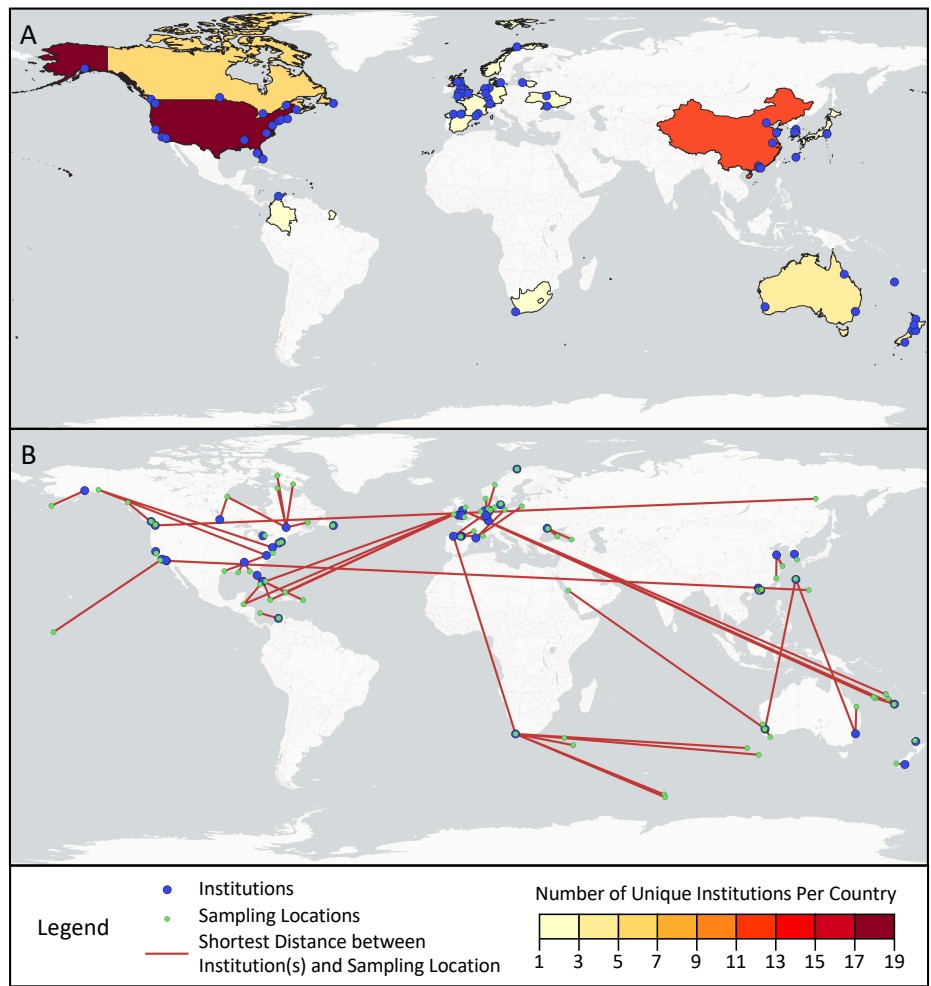

**Figure 4** **Geographic scope of sampled articles.** Map showing (A) the number of first and last author institutions located in each country and (B) the distances between the sampling locations and the one closest first or last author institution from a given paper, as visualized with red lines. Institutions from panel A that were not the closest institution to a sampling location from a given paper were omitted from panel B.

While most articles opted to use one of the above portals designated for DNA sequencing data, several articles decided instead to publish sequence data in more broad platforms emphasizing easily citable and open access data, including Dryad (10%) and Mendeley Data (1.7%). Finally, one article (1.7%) published sequence data *via* a platform, Qiita, specifically designed for the management and analysis of omics data.

Of the articles that published their sequence data, we further ensured that the link or accession number given successfully returned the data described. In several cases (10.9%), data were not available *via* the information given in the article (see Fig. 5). For each instance of missing data, we emailed the corresponding author(s) to ask whether they were aware that their data were not accessible, whether they had an explanation of the cause, and whether they knew how to address it. Of the five articles we emailed about, we

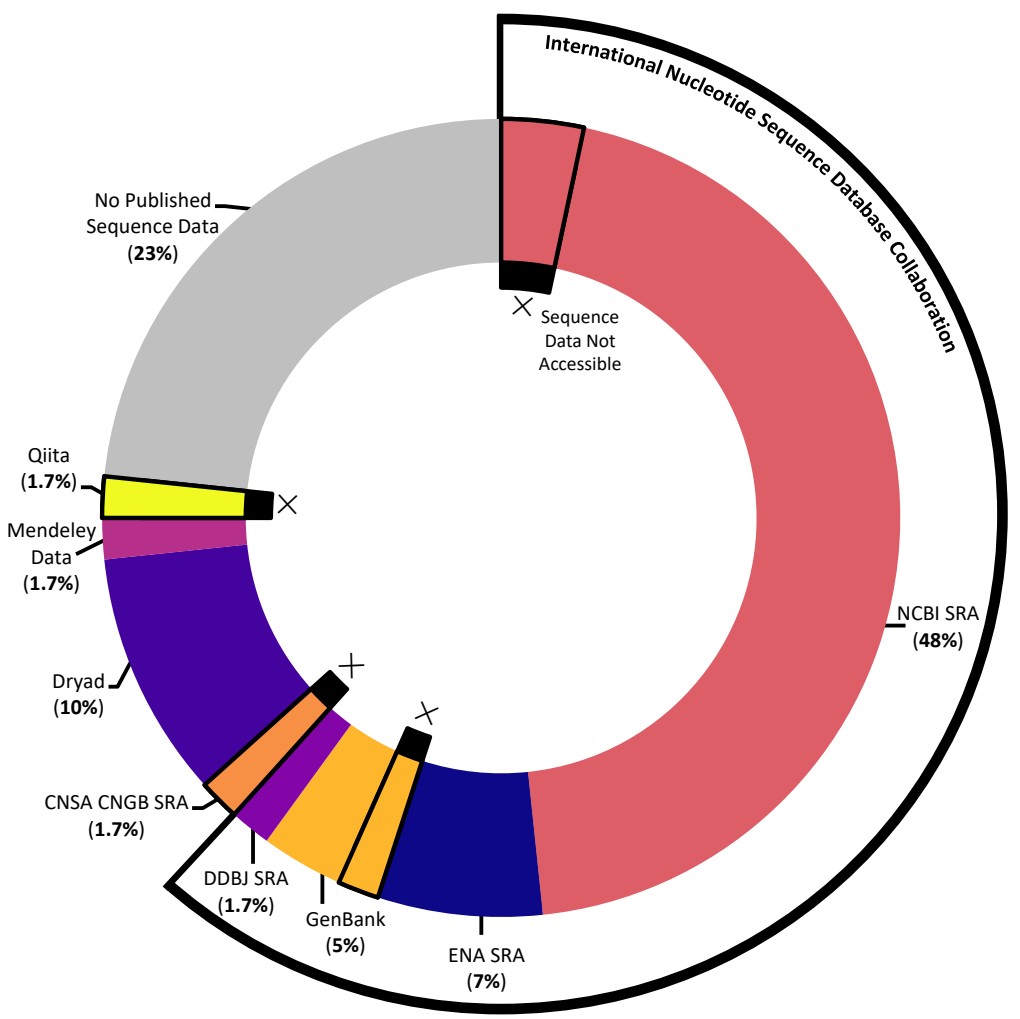

**Figure 5 Data storage platforms used across articles sampled.** Donut chart showing the percentage of articles in sample using different data storage platforms, including the majority of papers which utilized a platform under the International Nucleotide Sequence Database Collaboration. Donut sections outlined and marked with Xs denote papers where the link or accession number to access sequence data did not work at the time the systematic review was conducted.

received responses in four cases; these responses indicated some of the challenges involved in publishing DNA sequencing data. In several cases, authors described having issues when initially trying to upload the data that prevented them from successfully publishing their data as planned; in at least one instance, upon receiving our email, the author was able to remedy the situation. In one case, the paper had been published relatively recently, and the data submitted to the NCBI SRA had not yet reached its designated release date at the time we attempted to access it. And in one case, the author was confident that the sequence data had been accessible at one point and did not have a guess for what might have happened in the interim.
We also wanted to understand the link between open access publishing and sequence data availability: did a reader's ability to access underlying sequence data depend on how articles were published? Statistically, there was no significant relationship between open access status and whether sequence data were published ($\chi 2(1, N = 60) = .513$, $p = 0.473$); that is, articles that were published open access were not significantly more likely to make their data available. However, we found that *where* the information about published sequence data was located in the article made a difference for data accessibility when the articles themselves were not available open access. Articles were not consistent about where they placed the link or accession number that would lead readers to the underlying sequence data. Across the articles that published sequence data (76.7%), the link or accession number was sometimes in the main body text, including in the methods section and in the acknowledgements section. In other cases, the sequence data information was in a section external to the main body text, such as a data availability statement. Finally, some articles included the information in multiple locations. For open access articles, this location did not influence whether a reader could view the sequence data, since any viewer could see the whole article. But for the group of articles that did publish sequence data but were not available open access (15%), the location of the link or accession number determined whether any reader could still access that data. In that subset of articles, four articles provided information in an external section, such as a data availability statement or an ethics declaration; in those cases, we confirmed that the external sections were still visible even if the viewer did not have access to the full text of the article. The other five articles provided information about the sequence data publication in the methods section of the text, which would be behind a paywall for some viewers. Since so many marine eDNA metabarcoding studies do publish open access, these locational concerns only apply to a small portion of our sample. That being said, these findings indicated that over half of the subset of articles that were not available open access further had their sequence data obscured to readers who could not view the full text of the article.

## Metadata inclusion

Across the different metadata elements recorded (see Fig. 6), there was wide variation in the frequency with which they were included in articles, ranging from elements mentioned in 100% of articles sampled (such as sequencing instrument used and metabarcoding genetic marker) to those mentioned in 0% of articles sampled (such as wind conditions and precipitation at the location sampled). As shown in Fig. 6, the categories of metadata that had the highest average inclusion across the individual elements were filtration method (92%), sequencing (86%), and bioinformatics (78%). In contrast, the categories of metadata that had the lowest average inclusion across the individual elements were environmental conditions (8%) and controls (39%).

Across the 13 GBIF fields that corresponded directly with metadata elements assessed in this article (see asterisks in Fig. 6), only one article included 100% of the fields recommended for listing in GBIF. On average, articles included 65.6% of these GBIF fields. However, compliance increased when differentiated across the three levels of recommendation given for the GBIF fields: required, highly recommended, and recommended. Within the 13

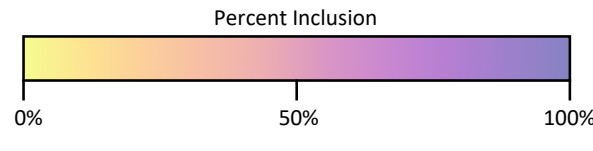

| Metadata Category (Average Across Elements) | Metadata Element | % Inclusion |
|---|---|---|
| Filtration (92%) | Filter Size & Type | 98% |
| | Preservation of Filter | 86% |
| Sequencing (86%) | Instrument | 100% |
| | Protocol* | 78% |
| | Read configuration* | 80% |
| Bioinformatics (78%) | Software/pipeline used | 100% |
| | Software version numbers | 80% |
| | Description of analysis | 98% |
| | Analysis scripts | 33% |
| Primers & PCR (78%) | Metabarcoding genetic marker* | 100% |
| | Subfragment of genetic marker* | 37% |
| | PCR Sample Replication | 73% |
| | PCR Purification Method | 80% |
| | Library Preparation | 87% |
| | Forward/Reverse Primers* | 73% |
| | Reference/citation for primers* | 91% |
| | Number of Cycles | 82% |
| | Length of cycles | 80% |
| Sampling Location (69%) | Country | 100% |
| | Province/State | 69% |
| | Sub-Location | 98% |
| | GPS Coordinates* | 66% |
| | Distance from Shore | 11% |
| Statistics (67%) | Software used | 87% |
| | Software version number | 79% |
| | Description of analysis | 88% |
| | Analysis scripts | 13% |
| Sampling (61%) | Depth | 72% |
| | Sample Volume | 97% |
| | Biological Replication | 45% |
| | Sample Container | 56% |
| | Water Sample Preservation | 59% |
| | Who Sampled | 37% |

| Metadata Category (Average Across Elements) | Metadata Element | % Inclusion |
|---|---|---|
| Extraction (59%) | Storage Time/Time Passed | 20% |
| | Kit/method | 98% |
| Data & Results (55%) | Particular OTUs/Sample* | 22% |
| | Total reads/sample* | 42% |
| | Level of taxonomic confidence* | 10% |
| | Reference database* | 100% |
| | Latin names or OTU identifier* | 100% |
| Sanitation (54%) | Field Decontamination | 56% |
| | Lab Decontamination | 55% |
| | Dedicated DNA-free space | 50% |
| Time of Sampling (53%) | Year | 92% |
| | Season | 18% |
| | Month | 90% |
| | Day* | 55% |
| | Time of Day | 12% |
| Controls (39%) | Field Blank | 50% |
| | Extraction Blank | 53% |
| | Inhibition Testing/Control | 5% |
| | Positive PCR control | 23% |
| | Negative PCR control | 65% |
| Environmental Conditions (8%) | Water Temperature | 20% |
| | Water pH | 7% |
| | Water Salinity | 17% |
| | Wind Conditions | 0% |
| | Precipitation | 0% |
| | UV Exposure | 0% |
| | Tidal Stage | 9% |

Percent Inclusion

0%    50%    100%

**Figure 6** **Metadata inclusion across categories & elements.** Table showing the metadata elements analyzed and the percentage of articles in the sample that included those elements. Elements are also grouped into overarching categories, with percentages denoting the average metadata inclusion across the individual elements in those categories. A color bar is used to shade the metadata elements and categories based on their percent inclusion. Elements derived from the GBIF recommendations for metabarcoding data are asterisked.

**Table 1 Metadata language used across sampled articles.** Table showing the number of papers using the word "metadata" to denote different categories of information.

| How term "metadata" was used | # of papers |
| --- | --- |
| Description of environmental/sampling conditions | 6 |
| Reference library information | 3 |
| Sequencing parameters | 2 |
| Sequence data | 2 |
| Information needed to cite another paper/script | 1 |

GBIF fields assessed, two were required, seven were highly recommended, and four were recommended. When assessing just required and highly recommended fields, three articles (5%) included all of the elements. When assessing just the required fields, compliance rose to 33 articles (55%).

The majority of articles (66.7%) did not publish any analysis scripts, and the remainder (33.3%) published scripts for bioinformatics and/or statistical analysis. For articles that published analysis scripts, the most common approach (50%) was to link to a GitHub repository containing some combination of scripts, data files, and protocols needed to reproduce analyses. Several articles (20%) used a combination of GitHub and supplementary information for storing analysis scripts, and two additional articles (10%) just used the supplementary information. Finally, there were a few platforms for publishing analysis scripts only used by one article each: FigShare, GitLab, NIOZ Data Portal, and Zenodo. As with sequencing data storage, these scripts were linked in various places in the article—*e.g.* the methods section, the supplementary information, or a data availability statement—with implications for whether a viewer could access the scripts if an article was not available open access.

### Metadata language

About one-quarter of articles (23.3%) used the word "metadata" somewhere in the text of the paper or its supplementary information. Across these articles, we observed its use in five main contexts (Table 1). Most commonly (six articles), "metadata" designated information about the environmental and sampling conditions. Alternatively, "metadata" sometimes referred to information about how reference libraries for taxonomic identification were constructed (three articles), information about sequencing parameters (two articles), and the original sequence data itself (two articles). In one case, "metadata" denoted the information needed to cite a set of analysis scripts (called "citation metadata").

## DISCUSSION

Our findings underscore some of the key challenges to designing best practices for marine eDNA metabarcoding, or eDNA work more broadly. At the same time, they also highlight some promising arenas.

## Key challenge #1: new and proliferating tools lacking a common context

Marine eDNA metabarcoding articles are diverse, finding homes in journals across disciplines and intersecting with a variety of different methodological approaches. In addition, the body of literature involving eDNA is growing quickly, as shown in Fig. 3. While this growth and lack of a narrow disciplinary or methodological context represent the widespread potential of eDNA tools, they also present challenges to standardization. For one, any attempts to develop best practices risk quickly being outdated, as a growing surge of new research challenges existing norms. At the same time, marine eDNA metabarcoding studies do not primarily fall in a single discipline or context, making it all the more difficult to find common ground for those best practices to build from. The objectives, contexts, vocabularies, and priorities of an environmental engineering team using metabarcoding to study eDNA fate and transport might be quite different from those of a conservation organization interested in tracking biodiversity over time. These projects could differ in practically all stages of the research process: from how they frame their research and choose their methods, to how they determine what metadata is important to save and share their results.

*Edwards et al.* (2011, p. 669) term this challenge "science friction": "the difficulties encountered when two scientific disciplines working on related problems try to interoperate". The challenge of science friction is not unique to marine eDNA metabarcoding. But this study focused only on a particular environment (marine) and approach (metabarcoding), and has already found wide divergences in context, metadata reporting, data storage, and more. Many efforts to develop standards focus on eDNA more broadly, including other environments and techniques; expanding in each dimension will only increase the potential for friction.

Other research programs have navigated these challenges before. In their analysis of long-term environmental research programs, *Edwards et al. (2011)* found that when operating across contexts and disciplines—researcher to database manager, geologist to ecologist—metadata existed not only as a product but also as a process: the communication and repair work needed to smooth over inevitable challenges storing and contextualizing complex datasets. In other words, especially in a context as diverse and ever-changing as marine eDNA metabarcoding, thinking about metadata, and best practices more broadly, solely as a checklist to be followed will never be fully sufficient. Trying to standardize how things are done can help reduce friction, certainly, but there will always be new users, new challenges, and new needs that emerge. Thus, efforts to standardize eDNA data storage and metadata practices ought to also consider how to foster what *Edwards et al. (2011)* call *metadata-as-process*, building infrastructure not only for storing information, but also to support ongoing communication, collaboration, and curation of that information. In the United States, there are burgeoning models for how to facilitate this communication. The eDNA Collaborative at the University of Washington is a new initiative designed, in part, "to build a network of researchers, sharing techniques and ideas" ("*The eDNA Collaborative*", n.d.). The National Workshop on Marine Environmental DNA, held first in 2018 at the Rockefeller University and again in 2022 at the Southern California Coastal

Water Research Program brought together diverse stakeholders—researchers, resource managers, industry representatives, and more—to accelerate the adoption of eDNA tools in new contexts (*Southern California Coastal Water Research Project, 2022*). Given the heterogeneity of marine eDNA metabarcoding research, continued efforts to connect researchers will be vital to support and complement the development of best practices.

However, our analysis also shows that these collaborative efforts may face a fundamental communication problem, with different stakeholders using distinct vocabularies and definitions. As one example, our analysis demonstrated that articles are delineating what constitutes metadata in divergent ways. Only about one-quarter (23.5%) of articles explicitly used the term "metadata" in the main article text or supplementary information, and when they did, they used it to refer to at least five distinct types of information (Table 1). This inconsistency is not surprising—most broadly construed as "data about data", the term "metadata" has come to encompass so much as to be functionally incomplete (*Sicilia, 2014*). But the finding that so few articles in the sample are using the word metadata, and that those that do are using it to mean so many distinct things, showcases that any efforts to build best practices across the diverse eDNA context may need to also build a shared language (*e.g.*, *Thompson et al., 2020*) for describing what core concepts—such as metadata—denote in the context of eDNA studies, or else risk being misinterpreted.

## Key challenge #2: restricted reusability: metadata & supplementary information limitations

Across the sample of articles analyzed, we found many instances where articles hindered the ability for the underpinning data to be reused. One key arena where this occurred was metadata reporting; as shown in Fig. 6, while some of the analyzed metadata elements were included in the majority of articles in the sample, others were never, or infrequently, included. In particular, the categories of environmental conditions and controls were the least frequently included of the metadata categories. These results both extend and complicate the findings of previous studies of eDNA metadata metrics. While they used a somewhat different set of metadata elements, in their systematic review of freshwater eDNA studies, *Nicholson et al. (2020)* similarly found that environmental conditions—wind conditions, precipitation, UV exposure, and pH—were some of the least reported metrics in the papers they analyzed. However, they found that positive and negative controls were described in 72.2% and 86.8% of papers, respectively (*Nicholson et al., 2020*). This may reflect differences in scope between their analysis and the one conducted here; while we focused exclusively on studies conducting eDNA metabarcoding, *Nicholson et al.* included papers using any eDNA methodology, so the majority of their sample of articles primarily employed qPCR (*2020*). While there have been efforts to standardize reporting of sequencing results (*Yilmaz et al., 2011*), these guidelines do not include information about controls; in contrast, the equivalent guidelines for qPCR experiments (MIQE, Minimum Information for Publication of Quantitative Real-Time PCR Experiments) include information about controls as essential to publication (*Bustin et al., 2009*). This discrepancy highlights the potential for misalignment between broad minimum information guidelines and the specific needs of the eDNA community. MIQE guidelines
may have helped institutionalize the reporting of controls in eDNA studies using qPCR (*e.g.*, *Abbott et al., 2021*), but the lack of similar guidance for sequencing results may be hindering the reporting of controls in eDNA studies employing metabarcoding. Navigating the intersection between best practices operating at different scales, and for different purposes—a documented tension (*Donaldson, Zegler-Poleska & Yarmey, 2020*)—will likely remain a challenge for efforts to develop eDNA-specific guidelines, especially given the disciplinary and methodological diversity of eDNA studies.

While our findings demonstrate that metadata standardization is needed, albeit challenging to design, they also show that such efforts are tough to operationalize once they exist. While several institutions have tried to standardize eDNA reporting requirements (*e.g.*, *Goldberg et al., 2016*; *California Water Quality Monitoring Council, 2023*; *Andersson et al., 2021*), we found that in one case—the GBIF metadata guidelines—only one article in our sample included all of the recommended GBIF metadata fields, although compliance rose to over 50% when only focusing on required metadata as opposed to highly recommended and recommended elements. Thus, the current metadata reporting norms across eDNA metabarcoding studies suggest that there will be significant barriers to implementing best practices and guidelines as they are introduced.

However, constraints to data reuse were not limited to metadata reporting. Only one-third of studies published their analysis scripts, either for bioinformatic processing or statistical analysis (Fig. 6), which would help new users analyze the underpinning data. And even among those studies that did publish analysis scripts of some kind, there was variation in their usability and accessibility. Some studies put their analysis scripts in supplementary information files without additional context—a list of Python functions or a minimally-commented R file. In other cases, studies linked to larger collections of code, on platforms including FigShare, GitHub, Zenodo, and the NIOZ Data Portal. But even when articles linked to these larger repositories, ease of use still varied widely; some collections of scripts were just a list of files with one basic read-me text, while others were formatted with significant instructions and documentation (and in some cases, with clearly demonstrated reuse potential as they were employed in several articles in our sample). Therefore, only a small portion of articles provided an easy, well-documented pathway for others to conduct similar analyses.

A similar variation in attention to reuse was found in supplementary information files. While most articles included some form of supplementary information, across a range of file types, we found many cases where choices made in presenting supplementary information reduced its accessibility and usability. For one, many articles with large numbers of supplementary files did not have sufficient documentation (in the article itself or in filenames) for determining what was included in each file, requiring readers to skim through many individual files to find the information they might be seeking. Additionally, there was sometimes a mismatch between the information being presented and its format—for example, lists of GPS coordinates or environmental variables in PDFs where they could not easily be extracted or copied, instead of Excel documents or text files. While these discrepancies are minor—and may often be a product of journal processes

rather than explicit choices by authors—they can add-up to much larger headaches for data reuse.

## Key challenge #3: democratizing eDNA data

Biodiversity research capacity is not well-distributed. Core biodiversity researchers are predominantly located in North America and Europe, even though regions in Africa, Asia, and South America have high biodiversity and more threatened species (*Tydecks et al., 2018*). Even when researchers in developing countries are included as collaborators on published papers, it is often in the context of tasks considered less scientifically valuable, like providing access to study sites and data; core tasks like contributing to study design and analysis are more frequently conducted at North American and European institutions (*Habel et al., 2014*). This sets up a problematic imbalance: locations most in need of increased capacity for biodiversity research are forced to rely on outside expertise (*Tydecks et al., 2018*). Another way of framing this imbalance is when outside researchers travel to locations just to extract data and "fail to invest in, fully partner with, or recognize local governance, capacity, expertise, and social structures", what is sometimes called "parachute", or "colonial", science (*De Vos & Schwartz, 2022*, p. 1).

While progress has been made in structurally addressing extractive biodiversity research, such as the Nagoya Protocol, a legal framework under the Convention on Biological Diversity meant to ensure the equitable sharing of benefits from using genetic resources, its applicability to digital sequence information remains contested (*Ambler et al., 2021*). In our analysis, we found that eDNA research may replicate biodiversity research inequities. Looking at first and last authors as a marker of the primary institution(s) involved in published studies, we found only 19 countries represented, with only one from Africa (South Africa) and one from Central and South America (Colombia). Similarly, nearly half (49%) of the 75 geographically distinct locations sampled by studies in our dataset are located in just six countries: the United States, Canada, China, Australia, Spain, and Norway. As shown in Fig. 4B, many papers in our study resulted from fieldwork conducted at a great distance from the primary institutions involved in the study, a potential marker of parachute science. Others have also documented similar geographic trends with eDNA studies. In a survey of eDNA articles that collected samples in Africa, *Von der Heyden (2022)* found few papers overall, sampling effort focused in a small number of countries, and one-third of papers with no authors from African institutions.

These geographical disparities in eDNA research coverage reflect a two-pronged problem: a dearth of studies covering sites outside North America, Europe, Australia, New Zealand, and East Asia, and a dearth of research institutions outside of those areas that have been able to conduct those studies. Thus, increasing accessibility of eDNA data means also ensuring that the places researched—and more importantly, the capacity for producing the research—are better distributed.

## Promising trend #1: consistency, and also creativity, in data storage

Looking at how articles in our sample opted to store their sequence data, when applicable, highlighted both consistency and creativity in data storage across marine

eDNA metabarcoding studies. For one, the majority of studies (80.4%) that published sequence data did so using a Sequence Read Archive (SRA) as part of the International Nucleotide Sequence Database Collaboration (INSDC). While the INSDC was formally consolidated under the current name in 2005, its origins date back to the late 1970s, establishing a multi-decadal trend toward considering the curation of DNA sequence information as a global project (*Stevens, 2018*). That so many eDNA metabarcoding studies are utilizing this global, well-established platform for storing sequence data demonstrates a promising step toward the interoperability of sequence data.

While the INSDC has enabled important standardization of data storage practices, the system also has shortcomings. Researchers we emailed about missing sequence data highlighted that submitting eDNA data to the INSDC can be difficult, and addressing errors is not user-friendly. More broadly, some researchers may also have concerns about free and unrestricted access to data, a core part of the INSDC's data-sharing policy (*Arita, Karsch-Mizrachi & Cochrane, 2021*). The 2021 State of Open Data report, a longitudinal survey of researcher perspectives on open data, found that concerns around open data are rising—from worries about the misuse of data to frustration around lack of credit or acknowledgement (*Simons et al., 2021*). Concern around lack of credit as a hindrance to data publication for reuse has also been documented among ecologists (*Zimmerman, 2007*). As outlined by *Berry et al. (2021)*, one limitation to better sharing of eDNA datasets in particular is that there are few formal incentives for researchers to spend the extra time and effort ensuring their datasets are as accessible as possible, leaving little to counteract potential concerns and hassles navigating systems like the INSDC.

While data storage consistency is an important way to work toward the interoperability of eDNA metabarcoding datasets, experimenting with new ways of archiving data in parallel can also help address shortcomings of the current dominant archiving system. Some articles in the sample utilized more broad open data platforms, like Dryad and Mendeley Data, which have several advantages to specialized sequence read archives. Both platforms allow files of any type (although with different maximum file sizes), allowing for easier inclusion of metadata, auxiliary datasets, analysis scripts, and more alongside sequence data files. Additionally, the published data can be linked to an associated published article, but it is also given a unique DOI, making it easier for new users of the dataset to cite it directly. However, because these platforms are so broad, the kind of verification and curation they can provide is necessarily quite basic. In the case of Dryad, most submissions are also subject to a data publishing charge. Beyond Dryad and Mendeley Data, one article in our sample utilized a more specific data platform, Qiita, designed for microbial studies to aid in data analysis and re-analysis. While the platform's specificity allows it to provide scaffolding for particular analytical processes, even with a verified account, we were unable to access the underpinning data with the information given in the article. As shown, no platform used by studies in our sample—whether broad or specific—worked perfectly for every application. But the current combination of approaches, with most studies defaulting to a consistent system but some exploring new options, seems like a promising path toward enabling interoperability while also making data as accessible, usable, and citable as possible.

## Promising trend #2: opening access

A recent large-scale analysis of DOI-assigned journal articles found that 27.9% were open access, and that articles published since 2000 are increasingly open access; in 2015, the last year analyzed, nearly 50% of articles were published open access (*Piwowar et al., 2018*). This analysis defined "open access" broadly as articles that are "free to read online, either on the publisher website or in an OA repository" (*Piwowar et al., 2018*). In our study, we only focused on a subset of that definition—those articles that were freely available directly on the publisher website—making the prevalence of eDNA metabarcoding articles we found to be open access particularly noteworthy. In our sample, 78% of articles were available openly from the publisher, and the four journals that published the most articles in our sample, representing 42% of the articles we analyzed—*Environmental DNA*, *Ecology and Evolution*, *Frontiers in Marine Science*, and *Scientific Reports*—all exclusively published open access articles. These findings suggest a much stronger culture of open access article publishing in the marine eDNA metabarcoding scholarly community than in academic scholarship at large.

However, making an article available open access does not guarantee that the sequence data underpinning the study are publicly available. We found that open access articles in our sample were no more likely to publish their sequence data than articles not published open access. Yet, the rate of sequence data publication did still mirror the open access publication rate; 76.7% of the articles in our sample indicated that they had published their sequence data. Thus, across both article and sequence data publication, we see a strong commitment to making scientific materials available in marine eDNA metabarcoding articles. It is important to note, however, that publishing underlying sequence data openly is not always the most appropriate or ethical choice. For example, results could be used to locate sensitive or endangered species, and there are growing concerns about ethics and data sovereignty of eDNA research conducted on Indigenous lands (*Handsley-Davis et al., 2021*).

That being said, the benefits of open access publishing can be numerous. For one, professionals, practitioners, and members of the public who might be interested in using environmental DNA tools increasingly have the option to pay to process samples *via* eDNA laboratories like Jonah Ventures, NatureMetrics, and Wilderlab, reducing the barrier to entry for conducting eDNA assessments. These potential eDNA users—such as resource managers, researchers at conservation NGOs, or community science organizers—may not have institutional access to peer-reviewed literature, making open access articles a potentially valuable venue for staying abreast of eDNA developments. Additionally, this larger potential audience can manifest concretely in how frequently the research is cited; one study found that open access articles "receive 18% more citations than otherwise expected" (*Piwowar et al., 2018*). *Berry et al. (2021)* detail how this occurred with eDNA open access data; a dataset from an eDNA study uploaded to an open-access biodiversity database, Atlas of Living Australia, has had thousands of records downloaded from it and has been cited seven times in other publications, an unusually high level of reuse. As this example shows, there is a particular interest in—and need for—open access data for biodiversity research and conservation science (*Fonseca & Benson, 2003*), especially to

ensure that there is equitable access to that knowledge in the developing world (*Gaikwad & Chavan, 2006*).

While many studies have lauded the importance of open access, it is important to note that the ability to publish open access articles often comes at a cost. In August 2022, the base article processing charges for the journals that published the most articles in our sample were $2,100 USD for *Environmental DNA* (*John Wiley & Sons, Inc., 2022a*), $2,200 USD for *Ecology and Evolution* (*John Wiley & Sons, Inc., 2022b*), $3,225 USD for *Frontiers in Marine Science* (*Frontiers Media S.A., 2022*), and $2,190 USD for *Scientific Reports* (*Scientific Reports, 2022*). While journals often offer small price reductions to members of professional societies, fee waivers and discounts for authors based in low- and middle-income countries, institutional payment agreements, and other mechanisms for reducing the burden of article processing charges, that many of the dominant journals publishing marine eDNA metabarcoding articles exclusively offer open access publishing means that scholars with limited funding not covered by other waivers and discounts may find it difficult to locate an attainable venue for their work.

That being said, our analysis has underscored that small changes in publishing decisions can have a big impact on the accessibility of underlying research data, even if articles themselves are not available open access. For the small group of articles (15%) in our sample that were not available open access but did publish their sequence data, more than half included the information about their sequence data in a location obscured by the paywall (such as the methods section) rather than in a location visible to anyone (such as a data availability statement). Knowing that the full text would not be available to everyone, those articles could have made a small adjustment—putting the link to their sequence data in a different part of the article—that would have made it easier for someone without access to the article to still view and use the data underpinning it. While it is certainly true that effectively reusing .FASTQ files could prove challenging without additional contextual details, some data storage platforms used by eDNA researchers, like Dryad, make it easier to include additional methods and information directly on the page where underlying data is hosted—a potential option for those interested in increasing data accessibility even if open access publishing is not a possibility.

## CONCLUSIONS

Marine eDNA metabarcoding research is making progress toward data FAIRness, but continued efforts are needed to ensure that data produced are sufficiently usable and accessible. While our systematic review found a trend toward open access article and data publication, we also highlighted many barriers to the interpretation and reuse of materials openly published, including a lack of common context and vocabulary across articles, missing metadata, and supplementary information limitations. Furthermore, across the articles we analyzed, sample collection and analysis were heavily concentrated in the United States, highlighting potential challenges to ensuring eDNA expertise and capacity is equitably distributed. Addressing many of these barriers will require significant efforts and coordination, but throughout our analysis, we also highlighted some areas

where decisions made by individual authors and journals could have an outsized influence on the discoverability and reusability of data, such as conscientious choices around supplementary information formats, data storage platforms used, and where information is placed in published articles.

As evidenced by our data storage results—where articles utilized INSDC platforms consistently but also experimented with other approaches—there is often a tension between standardization and innovation. Standards and guidelines can be important tools for helping increase data usability, as well as confidence in new methods, but they are imperfect tools; they can restrict new and creative approaches, are hard to universally adopt, and of particular importance for a fast-growing approach like eDNA, can be slow to adapt to a changing field. Thus, the aim of this systematic review is to provide additional data and perspectives to support continued conversations about how to make eDNA data more fair, accessible, interoperable, and reusable—that is to move *toward* best practices, not define them. To do that, our systematic review underscores that increased collaboration and coordination will be a vital underpinning to any efforts to structure how marine eDNA metabarcoding research ought to be done.

Our study also highlights many generative avenues for future research. For one, some of our findings—from the challenges of supplementary information formats to the instances of broken accession numbers—emerged first as barriers to our ability to systematically trace through published materials, rather than a priori extraction elements. Similarly, other more specific barriers to data usability might only become apparent when one attempts to actually reuse published data. Reproducibility studies—attempting to use published data and methods to reproduce findings from existing studies—might unveil other common challenges to data FAIRness, while also providing additional perspectives on what kinds of metadata are most vital for contextualizing results; plus, these can be an excellent tool for hands-on learning for newcomers to eDNA research. Additionally, bibliometric analysis could help elucidate how data practices vary across different communities of researchers; to what extent do studies that cite one another or emerge from researchers trained at the same institutions have similar trends in metadata reporting or data storage? Finally, like our analysis of the GBIF metadata fields, one could examine future proposed data standards or guidelines, to investigate retroactively whether published works could have complied with those standards; this process could be used as a benchmark for understanding whether new standards might be easy to adopt.

Incorporating new technologies into existing systems and frameworks is never easy. While this systematic review has characterized the many challenges to data usability and accessibility, more importantly, we hope it highlights the value of still striving toward, if not ever fully operationalizing, best practices for marine eDNA metabarcoding studies.

## ACKNOWLEDGEMENTS

Thanks to the staff of the Stanford Emmett Interdisciplinary Program in Environment & Resources, especially Ann Marie Pettigrew and Gabriela Magana, for all of their assistance. This study has been greatly enhanced by suggestions and feedback from many generous

interlocutors, including Susanna Theroux, Eily Andruszkiewicz Allan, Tyler Leeds, Matthew Mayernik, and Christine Borgman.

### Funding

This work was supported by an Emmett Interdisciplinary Program in Environment & Resources (E-IPER) Summer Research Grant and a Stanford School of Earth, Energy & Environmental Sciences McGee/Levorsen Research Grant. Meghan M. Shea is supported by a Stanford Interdisciplinary Graduate Fellowship and an E-IPER Sykes Family Fellowship. The funders had no role in study design, data collection and analysis, decision to publish, or preparation of the manuscript.

### Grant Disclosures

The following grant information was disclosed by the authors:
Emmett Interdisciplinary Program in Environment & Resources (E-IPER) Summer Research Grant.
Stanford School of Earth, Energy & Environmental Sciences, McGee/Levorsen Research Grant.
Stanford Interdisciplinary Graduate Fellowship and an E-IPER Sykes Family Fellowship.

### Competing Interests

The authors declare there are no competing interests.

### Author Contributions

- Meghan M. Shea conceived and designed the experiments, performed the experiments, analyzed the data, prepared figures and/or tables, authored or reviewed drafts of the article, and approved the final draft.
- Jacob Kuppermann performed the experiments, analyzed the data, prepared figures and/or tables, authored or reviewed drafts of the article, and approved the final draft.
- Megan P. Rogers performed the experiments, analyzed the data, prepared figures and/or tables, authored or reviewed drafts of the article, and approved the final draft.
- Dustin Summer Smith performed the experiments, analyzed the data, authored or reviewed drafts of the article, and approved the final draft.
- Paul Edwards conceived and designed the experiments, authored or reviewed drafts of the article, and approved the final draft.
- Alexandria B. Boehm conceived and designed the experiments, authored or reviewed drafts of the article, and approved the final draft.

### Data Availability

The data and materials used in the systematic review are available on Dryad, including a combined data sheet with all information extracted from articles, a record of each article screened and its inclusion decision, and a sample extraction data sheet.

The Dryad materials can be accessed through the following citation: Shea, Meghan et al. (2023), Systematic review of marine environmental DNA metabarcoding studies: Toward best practices for data usability and accessibility, Dryad, Dataset, https://doi.org/10.5061/dryad.95x69p8pd.

## Supplemental Information

Supplemental information for this article can be found online at http://dx.doi.org/10.7717/peerj.14993#supplemental-information.

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
