# Peer review of "Systematic review of marine environmental DNA metabarcoding studies: toward best practices for data usability and accessibility"

_PeerJ, doi:10.7717/peerj.14993_

## Round 0.1 · original submission · Minor Revisions

I agree with the reviewers that some revisions are in order for this manuscript to move forward, please attend to the reviewers' comments.

Reviewer 1 ·

Basic reporting

Please check comments in the attached document. Overall I find this review to be well-written, with most of my suggestions being technical rather than the fundamental structure of the review.

Comments as follows:
General comment: It doesn’t appear that the figures are numbered in the order in which they first appear in the paper.

General comment: It’s not clear to me where the figure legends are.

Lines 88-91: Shouldn’t this brief description of eDNA be moved to the Introduction, say at line 61?

Line 101: eDNA provides complementary species identification to traditional approaches based on morphotaxonomy?

Line 103: eDNA can be easier than other biodiversity survey techniques such as?

Line 107: eDNA has been used for a variety of purposes in marine environments such as?

Line 127: Commensurate is not quite the right word here, perhaps something like ‘A number of authors have identified that they had trouble integrating data across various project in a common framework, foregrounding the importance of systematic and comparable data collection and reporting procedures (Fediajevaite et al., 2021; Keck et al., 2022).’

Lines 142-145: Need a comma. Here’s a revision: ‘In response to reproducibility concerns, and a desire to increase confidence in eDNA results for use in regulatory and management contexts, there have been many efforts to provide standardized methodological and reporting guidance.’

Lines 162-164: Revision for clarity ‘While these efforts help guide researchers towards greater methodological standardization, the fact that so many parallel recommendations have been published highlights the challenge of adopting a single, universal set of practices.’

Lines 180-186: This sentence needs to be split into at least two.

Lines 186-188: Please elaborate a bit on how this software framework enables analyses across multiple eDNA projects.

Lines 195-199: Please elaborate for clarity ‘For example, the Genomics Standards Consortium developed minimum information about any (x) sequence (MIxS) as a general guide for metadata included with sequence (Yilmaz et al., 2011), and the Intergovernmental Oceanographic Commission’s Ocean Best Practices System developed the Minimum Information for an Omic Protocol (MIOP) as a guide for ocean-specific omics research (Samuel et al., 2021). ’

Line 199: Remove ‘And’.

Line 208: Add a comma after initiatives.

Lines 211-212: Revise ‘and none have focused on marine environments’

Line 212: Add comma after ‘yet’.

Line 216: Revise to ‘it centers on marine studies’

Lines 297-309 and Figure 6: Please clarify why these particular metadata categories were chosen (Were they just derived from GBIF, or somewhere else?). The categories which were included appear reasonable, but there are some categories which come to mind which would be useful which are not present. For example, for location metadata you included GPS coordinates but not spatial uncertainty. If someone is going to use location metadata to do something like extract remote sensing data at sampling locations then the scale of uncertainty, say 10m versus 200m, does matter.

Lines 305-397: Revise to ‘Across the first and last authors of the sampled articles, 78 different institutions were represented, ranging from universities and government agencies, to specialized centers and businesses (Figure 4A). ‘

Lines 427-429: Revise to ‘Other file formats found in supplementary information sections included various image data formats, as well as .FASTA and .FASTQ files associated with sequence data.’

Lines 444-447: Revise to ‘While most articles opted to use one of the above portals designated for DNA sequencing data, several articles decided instead to publish sequence data in open access and easily citable platforms, including Dryad (10%) and Mendeley Data (1.7%).’

Lines 447-448: It’s not clear to me what’s unique about Qiita if it’s just storing eDNA data.

Line 510: Those are three articles alone which meet the required and highly recommended categories, or three articles in addition to the 33?

Line 542: Please revise ‘Key Challenge #1: New And Proliferating Tools Lacking A Common Context ‘

Lines 567-571: Would it be possible to elaborate a bit on how this process of standardization works in geology? It would help with making sense of potential recommendations for the marine eDNA community.

Line 576: Please revise ‘metadata practices ought also consider how’ to ‘metadata practices ought to also consider how’.

Lines 634-638: Please add citations to this sentence as follows: ‘While several institutions have tried to standardize eDNA reporting requirements (citation), we found that in one case–the GBIF metadata guidelines–only a single article included all of the recommended GBIF metadata (citation for this article), although compliance rose to closer to 50% when only focusing on required metadata as opposed to highly recommended and recommended criteria.’

In the Discussion section ‘Key Challenge #3: Democratizing eDNA Data’ it would probably help to mention the Nagoya Protocol as at least an attempt at addressing the issues mentioned here.

Line 727: Was (2021) supposed to also include authors’ names?

In the Discussion section ‘Promising Trend #2: Opening Access’ it would help to elaborate more on efforts related to data sovereignty, particularly on how researchers have tried to address concerns related to issues such as geo-obscuring.

In the Discussion section ‘Promising Trend #2: Opening Access’, are there any attempts at funding open data access beyond discounts offered by publishers?

Line 856: I’m not sure how these conclusions emerged inadvertently since this was the result of a system literature review.

Line 873: Delete space in the phrase ‘it, a’.

Experimental design

No comment

Validity of the findings

No comment

Reviewer 2 ·

Basic reporting

The review in general is very basic and overly focused on a small set of the existing literature with the literature search having been conducted two years ago. The search methodology should be broadened to include other study system related terms. Many of the statements throughout the review are qualitative and should be made quantitative to avoid sensational writing. There needs to be clear objectives within the first section of the review. Overall, the review is far to general and limited in scope. While the intent is appreciated the current text seems far to narrative driven than academic.

Experimental design

Overly simple and needs to be expanded

Validity of the findings

Small sample size and limited scope

Additional comments

n/a

---

## Round 0.2 · accepted · Accept

This manuscript represents a significant effort in attempting to provide guidance towards studies using marine habitat eDNA assessment so that the impact might be better realized in the future.

·

Basic reporting

I was asked to review this article after a revision had been submitted, and in my opinion the manuscript used clear and unambiguous professional English throughout, sufficient background and context was provided, figures and tables were clear and understandable, and was clear and easy to understand the results.

Experimental design

I was asked to review this article after a revision had been submitted, and in my opinion the manuscript falls within the purview of the Aims and Scope of PeerJ, the research question is well-defined, relevant and meaningful, the analysis is rigorous, and methods are described in sufficient detail.

Validity of the findings

I was asked to review this article after a revision had been submitted, and in my opinion the manuscript and I found the findings of the study valid and conclusions well stated.

Additional comments

Great job!